# Developing a Curated Topic Model for
# COVID-19 Medical Research Literature

**Philip Resnik**
Linguistics/UMIACS
University of Maryland
resnik@umd.edu

**Katherine E. Goodman**
Epidemiology/Public Health
UMD School of Medicine
kgoodman@som.umaryland.edu

**Mike Moran**
SoloSegment
mike@solosegment.com

## Abstract

Topic models can facilitate search, navigation, and knowledge discovery in large document collections. However, automatic generation of topic models can produce results that fail to meet the needs of users. We advocate for a set of user-focused desiderata in topic modeling for the COVID-19 literature, and describe an effort in progress to develop a *curated* topic model for COVID-19 articles informed by subject matter expertise and the way medical researchers engage with medical literature.

## 1 Introduction

The language technology community has been responding in force to the coronavirus pandemic with sustained energy and creativity, and the medical research literature, facilitated by the CORD-19 dataset, is a major hub of activity (Wang et al., 2020). A crucial goal of all this technological effort is to support *non-technologist* users, namely the medical researchers, clinicians, and policy makers who are involved with the response to COVID-19. However, to date there has been little explicit discussion in the technology-development community about what functionalities will actually help these users in the trenches. Zhang et al. (2020) sum things up nicely when they write, "we don't actually know how our systems . . . can concretely contribute to efforts to tackle the ongoing pandemic until we receive guidance from real users who are engaged in those efforts . . . [The] challenge [is] how to build improved fire-fighting capabilities for tomorrow without bothering those who are trying to fight the fires that already raging in front of us".

In this paper we describe a cross-disciplinary collaborative effort that is intended to help close that gap by developing a *curated topic model* for COVID-19 medical research literature, informed both by subject matter expertise about the domain and by the way that medical researchers typically engage with medical literature. Our efforts make use of a human-in-the-loop platform for topic model development informed by the research literature on interactive topic modeling as well as by practical experience developing topic taxonomies for large scale document collections. The intent is not a new algorithmic contribution; rather, at its heart this is a position paper advocating for a needs-driven rather than technology-driven approach in NLP when tackling crucial real-world problems. We articulate and discuss what that looks like specifically for topic modeling of the COVID literature, and we offer for discussion our work in progress toward public release of a curated model that can be used downstream by the community as a resource and as a starting point for further work.

## 2 Desiderata

The core idea in topic modeling is to take in a collection of documents, discover a latent set of topics characteristic of the collection, and represent each document as a mixture of those topics. Typically each latent topic is itself represented as a distribution over the collection's vocabulary. Latent Dirichlet Allocation (Blei et al., 2003) is the ancestor of a whole variety of topic model variants, adding hierarchical topics, supervision, temporal structure, discrete covariates, and more, as well as more recent neural variations. In most applications of topic models, the topic-word distribution — i.e. the representation of each topic by its most probable $n$ words, for small $n$ — provides human consumers of the topic model with intuitions about what each topic is about, as a way of understanding the contents of the collection.

However, automatically discovered topic models are often subject to noise and poor quality (Boyd-Graber et al., 2014). In addition, the usual products

of topic modeling can be a poor fit for real-world use cases involving non-technical subject matter experts (SMEs) within a particular domain. From our own experience and discussions with medical researchers, we have identified a set of desiderata for topic modeling in the context of the COVID-19 literature.

- **SMEs are unlikely to be satisfied with purely bottom-up discovery of topics**. In order to obtain buy-in, topic models need to be substantially consistent with experts' pre-existing view of the world — even if one advantage of topic models is also to discover categories that were previously unknown.

- Related, **SMEs need to trust the resources they use**. In medicine there is a long tradition of painstakingly constructed category systems. Sometimes these involve boundaries that are *too* crisp; for example, over-rigidity in DSM diagnostic categories has motivated development of the multidimensional RDoC system in psychiatric research (Owen, 2014). However, the goal should be to augment rather than replace human training and expertise.

- **Topics need readable labels**. Reading a list of words to intuit an underlying topic or concept (or visualizing a set of such words) may be an acceptable starting point for model development but it will not suffice as an endpoint for SME model use.

- Related to the previous point, **documents matter**: the value of topics is closely tied to the documents that that have a high posterior probability for the topic. This contrasts strongly with the standard practice in NLP of focusing on the topic-word distributions (indeed, just the highest-probability words per topic) for interpretation, evaluation, and often use.

- **Valuable models are unlikely to result from one-shot analysis**. The process of developing a useful model involves looking at the model and the data, and then improving one, the other, or both.

- There is likely to be **high value in a special-purpose category system for COVID-19**, particularly for epidemiologists and specialists in clinical infectious diseases. General purpose categories (e.g. MeSH, Lipscomb, 2000) are unlikely to be sufficient for a rapidly emerging space with its own characteristics and concepts.

We find that these observations are largely in line with the principles for automated content analysis articulated by Grimmer and Stewart (2013) for the social sciences. In particular, they emphasize the role of automated analysis in expanding human capabilities, that models should be evaluated not intrinsically but in terms of their ability to support scientific goals, and they advocate strongly against the use of automated models without some form of validation. These observations are also consistent with Blei's (2014) discussion of "the craft of latent variable modeling" and his adaptation of "Box's loop" (Box, 1976) as an iterative picture of model development, although his discussion is focused much more strongly on the role of modelers than of subject matter experts.

We envision many uses for a curated topic model, but our driving use case involves the way many if not most medical researchers engage with the literature. A typical researcher is intimately familiar with Boolean searches in PubMed, including use of wildcards (e.g. `bacteri*`), quoted multi-word phrases (e.g. `''white blood cell''`), fielded searches (e.g. `aromatherapy[TIAB]` specifying to look in the articles' title, abstract, or keywords), and MeSH taxonomy headings and subheadings (e.g. `hypertension[mh] AND toxicity[sh]`). When reviewing search results, users will typically look at the article list and then, for a specific article, it is extremely common to drill down by looking at lists of articles surfaced by PubMed as *similar to* and/or *citing* the current article being looked at. Clicking through to other articles can be interleaved with new or refined Boolean searches.

The overall picture here is a very structured kind of search. In their day-to-day experience navigating the medical literature, it would be a fairly drastic shift from this way of doing things to typing in full-sentence questions in the style of question-answering systems or queries in the style of web search engines. When thinking about topic models, this motivates our thinking less in terms of, for example, improvements to ranking or visualizing topic-based clusters, and more in terms of discrete topic labels and how they could be introduced into the user's familiar, discrete experience, more anal-

ogous to MeSH terms. We would anticipate that one particularly valuable use of curated, labeled topical categories will be in helping researchers to overcome information overload when navigating results. For example, augmenting documents in the collection with discrete topic category labels would make it possible to organize related-article and citing-article lists into subcategories, and to include topical categories in follow-up searches.

## 3 Human-in-the-loop process

We are in the early stages of using an interactive, human-in-the-loop topic modeling platform to produce a curated topic model. The process we will be following is informed by prior experience using this platform to develop curated taxonomies for large document collections.[1]

We begin with preprocessing, including conventional steps such as tokenization (including identification of relevant multi-word expressions), lowercasing, removal of stopwords, and down-selection of the vocabulary to high-value words based on frequency and other statistical properties.[2] This is followed by creation of multiple initial topic models of differing granularities. We do not optimize the number $K$ of topics automatically, since doing so typically relies on automatic approximations to human judgment such as normalized pointwise mutual information (NPMI, Aletras and Stevenson, 2013; Lau et al., 2014). Rather, we will use human judgments directly by constructing models across a range of $K$ and assessing how promising each model is as a starting point, via a combination of qualitative assessment and by comparing human quality ratings for a random sample of topics as assigned by two independent SMEs.

Having selected an initial starting point, the human-in-the-loop process includes drilling down to better understand the model (including, for example, identifying documents that are highly representative of a given topic, or visualizing topic similarity), interleaved with human-feedback operations of the kind investigated by Lee et al. (2017)

and Smith et al. (2018). For example, they discuss within-topic feedback such as removing or adding words as good signals for a topic, as well as model-level feedback such as merging equivalent topics or removing topics that are too incoherent to be worth refining. Topics are also assigned labels manually, and labels can be revised at any time as the SME's understanding of a topic evolves. This feedback is provided in batch, and then the model is recomputed using the feedback to provide inductive bias. After the recalculation, the SME inspects the resulting model, and can either continue another iteration of refinement or designate the model as final. Typically two to four iterations of refinement using this process are sufficient to produce a high quality model.

Once a final model has been produced, we will evaluate its quality using multiple SMEs in epidemiology or clinical infectious diseases. We plan to use both subjective topic coherence ratings on a Likert scale (Aletras and Stevenson, 2013) and word intrusion (Chang et al., 2009), and to look not only at summary measures of agreement (e.g. correlation) but at specific sources of disagreement.

## 4 Preliminary analysis

We conducted a preliminary analysis of topic modeling using the titles and abstracts in the May 1, 2020 release of the CORD-19 dataset (metadata.csv).[3] This exploratory modeling used spaCy (spacy.io) for tokenization and identified phrase chunks in preprocessing as meaningful semantic units (Mimno, 2020). Although spaCy's phrase chunking performed very well, Mimno observes that "for text with lots of technical terms, a carefully curated list of multi-word terms can make a huge difference", and in subsequent exploration of initial models we will integrate COVID named entity resources to identify biomedical multi-word expressions, e.g. entities in Kroll et al. (2020).

Consistent with prior experience, we have found that *adding* multi-word tokens to a document's representation in preprocessing, rather than *replacing* unigrams, tends to yield topics that are more useful for SMEs, even though this does violence to the independence assumptions inherent in the typical topic model's generative process. For example, in a document containing the phrase *chronic headache*,

---

[1]As an example, a topic taxonomy was created for hundreds of thousands of documents in the content management system for a major national professional organization, producing human-readable topics that were integrated into faceted search on the organization's web site. Because we are using a commercial system and the desiderata and eventual curated model itself are the intended contributions, we provide an overview rather than sharing full technical details.

[2]We will use *word* to refer to both unigrams and multi-word tokens.

---

[3]We observe that use of titles and abstracts, rather than full text, is consistent with how article similarity is calculated in PubMed (PubMed Help, 2020).

we would include all of *chronic_headache*, *chronic*, and *headache* as tokens. This can introduce some redundancy, but it also allows the model itself to determine, for a given topic, whether (for example) *chronic_headache* or *headache* is the appropriate level of abstraction for the topic in the context of other terms.

In our preliminary analysis, we constructed and inspected initial models with $K = 50, 100, 150$ topics. On drilling down into the models, it became apparent that the collection contains a high proportion of articles that had been retrieved by the CORD-19 dataset search terms but were not directly relevant to COVID-19. On further inspection, we have come to the conclusion that from an epidemiological and public health perspective, a cleaner and more valuable set of curated topics for the COVID-19 research literature is likely to be created if the inclusion criteria for the document collection are stricter, limiting to articles sufficiently "about" COVID-19 by creating a filter based on the inclusion search terms used in creating the CORD-19 dataset (Wang et al., 2020), further tuned to focus on *novel* (2019) coronavirus, not all previous coronavirus research in general.[4]

In addition, as is common in English-dominated but multi-language collections, we have found in our preliminary analyses that modeling tends to produce topics aggregating words in non-English languages. As a simple way to address this issue, we have determined that a simple heuristic filter for English documents works well: requiring that the title or abstract contain at least one of the most common words in English.[5] Of the roughly 3000 items filtered using this heuristic (about 5,500 in the June 7 release), only a tiny number include abstracts in English. In addition to non-English content, the heuristic picks out some articles where a title is provided but the abstract is empty; technically speaking these are false positives for the

language filter, but arguably they represent a subset of the collection that is likely to offer less value than titles plus abstracts.

Our initial impressions of topic granularity suggest that $K = 50$ topics is going to be too coarse grained for a collection of this size. This is consistent with our prior experience with human-in-the-loop modeling for collections with tens of thousands of documents, where we have found that 100-150 topics was a good initial starting point. For the interactive refinement process, it makes sense to err in the direction of fine-grained topics in the initial model, since our refinement process makes it easy to whittle down the set by identifying, sharpening, and merging topics that cover the same conceptual territory and by removing manifestly incoherent topics.

To provide an impression of the topic model obtained in our initial process we show three randomly chosen topics from the 100-topic model for the May 1 release, noting that initial model results are expected to change once we implement the stricter inclusion criteria and medical entity processing (and of course reiterating our own prior cautions about looking only at highly ranked topic words):

- traditional_chinese extract plant plants extracts compounds activities medicine biological_activities tcm chinese natural_products chinese_medicine natural glycyrrhizin flavonoids quercetin traditional_chinese_medicine action inflammatory

- mental_health anxiety life stress depression individuals self post women physical disorder psychological chronic_diseases scores scale exercise quality status health_status risk_factors

- membrane endoplasmic_reticulum plasma_membrane ifitm3 membrane_proteins membrane_protein cell_surface golgi transmembrane_domain golgi_complex golgi_apparatus transport domain membranes channel terminal transmembrane_protein transmembrane release ion_channel

In general, informal SME judgment lends us some confidence that the preliminary choices made thus far have put us on the right track.

## 5 Conclusions

We have argued for the importance and utility of a curated topic model for COVID-19 medical research literature, proposed criteria that such modeling should satisfy, and we have provided a preliminary sketch of how we are planning to construct a curated model meeting those criteria.

---

[4]The reasoning here is that, within a given "budget" $K$, it is important to provide sufficient granularity for COVID-19-specific topics. Now that we have progressed beyond the earliest days of the epidemic, well-developed subareas are emerging in the COVID-19 medical research literature, but casting the net broadly enough to include all prior coronavirus discussion appears to drown out themes related to the *novel* coronavirus more than it helps capture themes that are relevant to it; for example, most of those studies are *in vitro* with at best a tangential relationship to our human coronavirus. Retaining the SARS- and MERS-related search terms ensures that relevant pre-2019 literature related to human coronaviruses is still included.

[5]Specifically *the*, *be*, *am*, *are*, *is*, *was*, *were*, *being*, *been*, *to*, *of*, *and*, *a*, *an*, *in*, *that*, *have*, *i*, *it*, *for*, *not*, *on*, and *with*.

This is work in progress, but preliminary analysis suggests that initial topic models are high quality, we have identified several steps for improving initial automatic modeling, and our process of human-in-the-loop refinement is designed to further refine and curate a model whose content will be driven both bottom-up by the data and top-down by subject matter expertise.

One important take-away from our preliminary investigation is the importance of careful sample selection as guided by the end goals. This consideration is entirely *de rigeur* in content analysis as practiced in other disciplines (Smith, 2000), but within NLP there is often a tendency to work with "found data", using it in its entirety. We find that for the goals we are pursuing here, more data is not necessarily better data.

At the end of our curated model development process, we will make publicly available topic-word distributions with corresponding labels, document-topic distributions for the input documents, the vocabulary, and a script that enables pre-processing consistent with ours for new documents. This will provide the community with materials needed to visualize the topics and documents, to manually organize topics into a hierarchy, to compute topic posteriors for new documents, to treat a document's most-prevalent topic(s) as discrete labels, and to integrate topic-labeling of documents into users' experience in search engines.

More generally, one of our key aims is for this curated model to serve as a useful starting point for further model development by the community, e.g. using our resource to construct informative priors in development of further models. This may help others in bootstrapping work on this dataset using generative models, but it also represents one reasonable answer to the question of how to deal with a literature that is rapidly and continuously evolving, namely starting with the curated model as $M_0$ and then periodically performing inference on an updated document collection at time $t$ using $M_{t-1}$ to define a prior. Related work along these lines includes, among others, lexical "seeding" of topics (Lu et al., 2011; Jagarlamudi et al., 2012), treating the literature as a streaming document collection (Yao et al., 2009), and dynamic topic models (Blei and Lafferty, 2006). The best way to introduce additional curation over time remains an important question for future work.

The seriousness and urgency of COVID-19 re-quires the creation of new knowledge on an enormous scale. Paradoxically, though, the deluge of work on this subject, producing more and more knowledge, is itself an obstacle to progress. Clinicians, medical researchers, and policy makers cannot read everything — they need better ways of making sense of what's out there, organizing it, and navigating in the directions that will help them. Technology that is focused on their needs can help.

## Conflict Statement

Mike Moran is Chief Product Officer of SoloSegment, a private company that is commercializing the interactive topic modeling platform we use. Philip Resnik is an advisor to SoloSegment.

## Acknowledgments

We are very grateful to the reviewers for thoughtful and insightful comments. This paper has benefitted from helpful discussions with Steven Bedrick, Dina Demner-Fushman, Lucy Lu Wang, Kyle Lo, and Sebastian Kohlmeier, Kuansan Wang and Iris Shen, Christopher Pilcher MD, and Amy Baxter MD. The authors take full responsibility for all opinions expressed here. This work has been supported in part by Amazon's initiative to accelerate COVID-19 research and by the National Science Foundation under Grant No. 2031736.

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
