# OpenReview forum: "Developing a Curated Topic Model for COVID-19 Medical Research Literature"
_EMNLP/2020/Workshop/NLP-COVID — NLP-COVID19-EMNLP Poster_

### Official Review · AnonReviewer2 · 2020-09-11
**Highly preliminary, but with a compelling analysis of desiderata.**

**Rating:** 4
**Confidence:** 3

**Review:**

This paper presents preliminary work towards a topic model for use by Covid-19 researchers, being developed with a human-in-the-loop process. It explains the needs of the subject matter experts (SMEs) who are to use the model, and how fully automatic topic models do not meet their needs. They present the work they have done so far in constructing automatic topic models as a starting point, and describe their plans for the involvement of humans.

Reasons to accept: Their analysis of the desiderata for topic models, and their anticipated uses, is extensive, highly compelling, in line with my own experiences of attempting to deploy topic models and similar systems, and likely to stimulate interesting discussions at the workshop. The initial NLP work looks reasonable, although the details there are vague. The final resource is likely to be of value to NLP workers and SMEs alike.

Reasons to reject: Their description of the plans human involvement is frustratingly brief and vague; the most useful part seems to be a pair of references to other group's work in 2017 and 2018. I get the impression that plans have not been finalised. The vaugeness is also a problem with their descriptions of the NLP work already conducted - the descriptions are a long way from providing reproducibility. Finally, the description of the specific challenges of working with Covid-19 research, rather than biomedical research in general, is minimal.

Overall: This work is highly preliminary, and thus lacking in detail, and as such, in my opinion, publication of this work at this stage would be premature. However, much depends on what the organisers aims for the workshop are; it would make for a stimulating presentation and lively discussion on the day, and if the organisers prioritise this above making a long-term contribution to the literature, then maybe it should be accepted.

---

### Official Review · AnonReviewer1 · 2020-09-24
**A good reflection on bringing topic models to medical experts, but lacking in detail**

**Rating:** 6
**Confidence:** 4

**Review:**


This paper presents a description of ongoing work involving curated topic modeling for improved exploration of COVID-19 literature. The idea is based on including humans in the loop, so that the initial topic model can be better fit to experts’ needs. The authors reflect on a list of desiderata that would make the use of topic models more acceptable to medical practitioners, such as the use of readable labels and structured search. Then, they discuss their ongoing efforts in this direction.

The paper is clearly written and a pleasure to read. I especially liked the discussion of the desiderata that would enable easier adoption by medical experts, and the description of the intended use where the topic labels would be used in a way similar to MeSH terms.

The actual work in progress is presented in broad terms only (with a few technical details for the more trivial parts such as preprocessing), which is in my opinion the biggest shortcoming of the paper. I am wondering, for example, how the model is precisely recomputed after the user has provided feedback (incorporation of label information, deletion of words, removal of topics etc.). And, how are the topic labels assigned? More generally, since the document collections are ever growing, how do the authors plan to adapt the topic models when the new data comes in?

I would like to draw the attention of the authors to a related paper at this workshop, which also studies medical multi-word expressions in topic models: https://openreview.net/forum?id=c-TkXmZC-Yk (Improved Topic Representations of Medical Documents to Assist COVID-19 Literature Exploration). The paper also contains other ideas about refining topic models for COVID-19.

Editing remarks:

When introducing the last desideratum, the authors mention that special purpose category systems are needed for COVID-19, but the link to topic models was initially opaque to me. It  became clear only at the end of the section.

The acronyms DSM and RDoC may be unfamiliar to the reader.


Overall: Although I would hesitate to accept the paper for its empirical contributions, I do think the paper could lead to interesting discussion at the workshop due to its contemplating nature about the inclusion of an NLP component into medical literature search.

---

### Official Review · AnonReviewer3 · 2020-09-24
**A potentially important contribution but still too initial to evaluate its merits**

**Rating:** 5
**Confidence:** 2

**Review:**

**Core review**

This paper proposes a methodology for curating a topic model from CORD-19 documents to improve downstream NLP tasks in this domain. Authors argue about the importance of curated vs fully automatic models and outline a set of requirements that a high-quality topic model should have, in order to be useful for end-users. The paper also provides some preliminary analysis on the current state of this endeavour, but details are still too superficial to evaluate if the quality of the curated topic model is indeed a significant factor for improving other downstream tasks like QA or IR.

I consider this paper could spark an interesting discussion in the workshop given the inter-disciplinary nature of the work, and this type of discussion is very necessary if the biomedical NLP research community pretends to produce results that are usable in practical scenarios. My primary concern is that the results are still very initial to evaluate the feasibility of the proposal or estimate the value that this resource could provide. The authors appear to have previous experience in this type of task, which is an argument in favour of their success. However, I would have appreciated more details that could help quantify the expected returns of this effort.

**Reasons to accept:** A very interesting and important discussion that can potentially benefit the research community and a very good exposition of the necessity of this resource and its requirements.

**Reason to reject:** Preliminary results are too superficial to allow evaluating the feasibility

---

### Author Response · Authors · 2020-09-27
**Author response**


Many thanks for the careful reading of the paper and the thoughtful consideration!

As noted in the introduction, our main aim here is not a novel technical contribution. Rather, it is to get what we believe to be some highly relevant issues worth considering out in front of the community for discussion, along with our own proposals for how it would make sense to tackle them and what we have done so far. We hope for feedback, but, more important, we hope it will be an opportunity to get other people thinking and talking about these issues, whether they wind up being on board with our approach or are inspired to take things in a different direction. That's a conversation where it's best for discussion to take place before the proposed resource has been completed, not after. So... in some sense the question is whether the preliminary nature of the work is a bug or a feature. :)

As another way of putting it, looking back a bit (or more than a bit!) in history, the SIGLEX 1997 workshop paper by Resnik and Yarowsky was in essence a position paper about how the community should approach evaluation of word sense disambiguation, accompanied by technically oriented proposals for discussion. Energetic workshop discussion led to the SIGLEX community's creation of the SENSEVAL evaluations, which evolved into SemEval. The community did not adopt all aspects of what was proposed, but the paper inspired a broad consensus on what was needed and provided an initial framework for discussion. At its most ambitious, this submission is going for something similar: it is, at its heart, a position paper, offering a set of issues we believe worthy of discussion and our own perspective and presenting preliminary efforts there as a framework for discussion.


Looking at comments specifically by AnonReviewer1, we won't be providing our own model recalculation details (see Footnote 1), but Smith et al. (2018) discuss methods for that. We will, however, be providing full details on data selection and NLP preprocessing scripts and their documentation (see Conclusions). Label assignment for topics is done by the subject matter expert (SME); in our experience, initial assignment of labels provides useful conceptual hooks, and then these labels are updated by the SME as their holistic picture of the topic set evolves.

The question the reviewer raises of adapting the models as the document base grows is quite important, and it's arguably more urgent for the COVID-19 literature than elsewhere. The longitudinal view of documents is largely under-emphasized in the literature on topic models (with some exceptions, e.g. in the line of work on dynamic topic models).  We would love to get the community talking seriously about this issue (see discussion of our overall aims, above) and we believe that our releasing a curated model is a decent starting point, since this could then be used as inductive bias for new modeling integrating updated data, which could then produce new models released as a starting point, etc.

AnonReviewer2 observes that the challenges we identify could be seen as issues for topic modeling in biomedical research in general, not just specific challenges for working with COVID-19 literature.  We agree -- though to us this servers to reinforce the breadth of the disconnect between the usual (if not absolutely universal) technology-driven approach that's taken in our community and the more use-case-driven emphasis we are advocating.  That said, COVID-19 does present or at least highlight some particular challenges. One, as noted by AnonReview1 above, is the dynamic nature of this rapidly growing literature and what to do about it. Related, our emphasis on the importance of data selection comes about as a result of the observation that a fixed number of topics provides a limited "budget", so to speak, and including the large body of pre-COVID-19 literature in CORD-19 turns out (according to our SME) to allocate too much of that budget to non-COVID-19 concepts.

AnonReviewer3, in discussing the preliminary nature of the paper, frames the key limitation of the submission in terms of evaluating feasibility and assessing expected returns. Although we are optimistic about feasibility and the eventual outcome for our own approach, this is a fair point. We have not submitted completed results, and there are no guarantees.  But, to return again to our primary point above, we believe that whether or not our own approach is exactly the right one, the *goal* is the right one, and that it is well worth the workshop-level discussion.  If someone else picks up on the idea that topic model development for COVID-19 should be driven by the needs of the medical researchers, clinicians, and policy makers who are in the trenches on the response to COVID-19, and they get somewhere with it better or faster than us, we will be the first to celebrate.